# Productive wetlands restored for carbon sequestration quickly become net $CO_2$ sinks with site-level factors driving uptake variability

Alex C. Valach[1¤a]*, Kuno Kasak[1,2], Kyle S. Hemes[1,3], Tyler L. Anthony[1], Iryna Dronova[1,4], Sophie Taddeo[4¤b], Whendee L. Silver[1], Daphne Szutu[1], Joseph Verfaillie[1], Dennis D. Baldocchi[1]

1 Ecosystem Science Division, Department of Environmental Science, Policy and Management, University of California, Berkeley, CA, United States of America, 2 Department of Geography, Institute of Ecology and Earth Sciences, University of Tartu, Tartu, Estonia, 3 Woods Institute for the Environment, Stanford University, Stanford, CA, United States of America, 4 Department of Landscape Architecture and Environmental Planning, University of California, Berkeley, CA, United States of America

¤a Current address: Agroecology and Environment Division, Agroscope, Zurich, Switzerland
¤b Current address: Negaunee Institute for Plant Conservation Science and Action, Chicago Botanic Garden, Chicago, IL, United States of America
* a.c.valach@gmail.com

**Data Availability Statement:** All sites used in this analysis are part of the AmeriFlux network, with data available at http://ameriflux.lbl.gov/ with

## Abstract

Inundated wetlands can potentially sequester substantial amounts of soil carbon (C) over the long-term because of slow decomposition and high primary productivity, particularly in climates with long growing seasons. Restoring such wetlands may provide one of several effective negative emission technologies to remove atmospheric $CO_2$ and mitigate climate change. However, there remains considerable uncertainty whether these heterogeneous ecotones are consistent net C sinks and to what degree restoration and management methods affect C sequestration. Since wetland C dynamics are largely driven by climate, it is difficult to draw comparisons across regions. With many restored wetlands having different functional outcomes, we need to better understand the importance of site-specific conditions and how they change over time. We report on 21 site-years of C fluxes using eddy covariance measurements from five restored fresh to brackish wetlands in a Mediterranean climate. The wetlands ranged from 3 to 23 years after restoration and showed that several factors related to restoration methods and site conditions altered the magnitude of C sequestration by affecting vegetation cover and structure. Vegetation established within two years of re-flooding but followed different trajectories depending on design aspects, such as bathymetry-determined water levels, planting methods, and soil nutrients. A minimum of 55% vegetation cover was needed to become a net C sink, which most wetlands achieved once vegetation was established. Established wetlands had a high C sequestration efficiency (i.e. the ratio of net to gross ecosystem productivity) comparable to upland ecosystems but varied between years undergoing boom-bust growth cycles and C uptake strength was susceptible to disturbance events. We highlight the large C sequestration potential of productive inundated marshes, aided by restoration design and management targeted to maximise vegetation extent and minimise disturbance. These findings have important implications for wetland restoration, policy, and management practitioners.

designations US-Tw1, US-Tw4, US-Tw5, US-Myb, and US-Sne.

**Funding:** This work was supported in the form of funding by the California Department of Water Resources through a contract from the California Department of Fish and Wildlife and the United States Department of Agriculture (NIFA grant #2011-67003-30371) awarded to DDB. Funding for the AmeriFlux core sites was provided by the U.S. Department of Energy's Office of Science (AmeriFlux contract #7079856) and the aerial images and footprint mapping was funded by the Delta Science Program grant #R/SF-52 awarded to DDB and KSH. KK was supported by the Estonian Research Council grant No. PSG631 and by the Baltic-American Freedom Foundation Research Scholar program. KSH, ST and TLA were supported by the California Sea Grant Delta Science Fellowship (programs R/SF-70, R/SF-71 and R/SF-89 and grant no. 2271 and 5298). McIntire Stennis grant CA- B-ECO-7673-MS awarded to WLS partially supported this work. This work uses data and processing services provided by the OpenTopography Facility with support from the National Science Foundation under NSF Award Numbers 1948997, 1948994 & 1948857.

**Competing interests:** The authors declare no competing interest.

# 1 Introduction

Peat-dominated ecosystems contain the largest global terrestrial soil carbon (C) stores [1–3], with freshwater marshes accounting for almost 30% of C stocks while only covering around 5–8% of the land surface area [4–6]. There is growing interest in wetlands for their capacity to store C given the long residence time. This is achieved through anaerobic conditions which protect existing soil C, while vegetation continues to sequester atmospheric carbon dioxide (CO$_2$). The wetland acts as a negative emission technology, which helps to mitigate climate change [7–9]. Although anaerobic conditions reduce C loss from decomposition, these conditions, especially in inundated fresh to brackish wetlands, produce methane (CH$_4$) and possibly nitrous oxide (N$_2$O) emissions [10]. These are both strong GHGs, which can significantly increase the global warming potential of wetlands over decadal time periods reducing the climate mitigation benefits of restoration [11–13]. However, a previous study of these wetlands showed that in some years they were immediate net GHG sinks, because the high net C uptake offset the climate forcing of the CH$_4$ emissions [14], even without accounting for N$_2$O uptake, which was previously found at these sites [9]. Therefore, understanding the drivers of C uptake in similarly productive freshwater wetlands is necessary to better identify and manage them as natural climate solutions. Despite this, many wetlands are being degraded by human activities or threatened by climate change [15,16], causing large C and greenhouse gas (GHG) emissions and depleting soil C stocks [17]. Many short-term datasets (<5 years) of wetland C fluxes show high C uptake variability with wetlands ranging from being strong C sinks to C sources [e.g. 18–21]. This translates into considerable uncertainty about whether restoring wetlands offers an effective negative emission technology in the long-term and a need to identify which conditions improve this functionality [6,22–26].

For wetlands to effectively sequester C, net C uptake must largely remain stable in the face of both small-scale and large-scale perturbations, from local disturbance to future environmental equilibria [22,27–31]. While climate factors are an important control on wetland plant productivity and ecosystem respiration [32,33], they do not fully explain the high interannual variability [34], especially between wetland types and sites [15,35]. Evidence suggests that site-specific factors including restoration design, patterns in disturbance and succession, past land use and the effects of management practices, such as water level manipulations may play a large role in the annual net C balance of wetlands [36,37]. There are only few long-term datasets of ecosystem-level C uptake measurements [27,28], such as presented here, which are necessary to better understand the lasting net C uptake strength and sink stability. Exploring site-specific controls requires investigating multiple wetlands within the same climate under a range of restoration methods and site conditions with a suite of additional data on hydrology, soil, and vegetation [15,38,39].

The eddy covariance method allows for direct measurements of net ecosystem exchange (NEE) over an integrated ecosystem-scale area and generally assumes that the sources are homogeneous and representative of the whole site. While being micrometeorologically ideal in terms of their flatness, wetlands have high spatial complexity with regards to sources and sinks which recent publications showed is often underrepresented in many flux studies. Although there is a growing effort to quantify and account for spatial heterogeneity within the measurement area of the flux, i.e. the flux footprint [40], only few C flux datasets currently provide the wide array of supplementary measurements needed to interpret the spatial heterogeneity in more detail. This makes it difficult to connect spatial patterns, such as specific restoration and management strategies to ecosystem-level fluxes as restored ecosystems develop over time. Here we used an interdisciplinary approach that combined 21 site-years of eddy covariance

measurements from five restored marshes with data on soil, water, vegetation, and meteorological conditions to better understand spatial and temporal variations in C fluxes.

The Sacramento-San Joaquin Delta (hereafter the Delta) in California, USA, represents an ideal case study for the re-establishment of wetlands following drainage for intensive agriculture [41,42]. There are several wetland restoration projects in the Delta, which provide a unique opportunity to investigate how site-specific factors like land cover and vegetation development affected yearly C budgets following restoration, as these contrasting wetlands experienced the same climatic conditions. In the Delta, the previous agricultural land-uses and hydrological modifications have changed the fundamental conditions of the wetlands, therefore we use the term 'restored' to describe the re-establishment of constructed marshes, albeit under artificial conditions [43–45].

The continuous NEE measurements provide the difference between the capture of atmospheric $CO_2$ from gross ecosystem productivity (GEP) via photosynthesis minus any losses to the atmosphere from autotrophic and heterotrophic ecosystem respiration. Lateral transport of dissolved and particulate C is not captured by this method, but although it has been shown to constitute a major component to C loss in natural systems [33], at these sites it is assumed to be negligible due to the limited outflow from the wetlands [46]. The C and GHG budgets of these wetlands including an assessment of the $CH_4$ emission impacts are discussed in detail elsewhere [14].

We tested the null hypothesis that these wetlands, being located within a few kilometres of one another with similar vegetation, would converge in terms of their structure and function. We investigated 1) how vegetation cover and structure developed over time, as well as 2) the relationship of different land cover designs and vegetation types with annual net C uptake across the five restored marshes over a 10-year period.

## 2 Methods

All sites in this study (i.e. Sherman and Twitchell Islands) are owned by the State of California and access was provided by the state's Department of Water Resources. The field studies did not involve endangered or protected species.

### 2.1 Study sites

**2.1.1 The Sacramento-San Joaquin Delta.** In the mid-19th century, the deep peat soils of the Delta wetlands were drained for agricultural purposes, exposing the C-rich soils to oxygen. Microbial oxidation resulted in an estimated loss of 83–100 Tg C [41] and together with peat compaction caused widespread land subsidence with some areas now 10 m below sea level [43,47]. A network of levees maintains the drained islands [45] but growing hydrostatic pressure from continued subsidence and sea level rise increases the risk of levee failure [43,47]. Levee breaks are costly and oftentimes irreversible, endangering much of California's freshwater system centred in the Delta [48], which provides over 27 million people with drinking water [46] and agricultural irrigation [45]. Wetland restoration projects in the Delta primarily aim to stabilise levees and reverse land subsidence by accreting soil [49], which also has the potential to mitigate climate change by sequestering C and thereby provide additional funding from C markets to prolong the economic viability of restoration projects [50,51]. Many projects in the Delta advocate for co-equal goals [52–54], including C sequestration, flood protection, water quality and supply, biodiversity habitat, and outdoor recreation [55]. However, different restoration goals can produce trade-offs when restoration practices conflict with each other [56–59], therefore we solely focus on C sequestration in this study.

**2.1.2 Site histories.**   West Pond, East Pond, and East End located on Twitchell Island and Mayberry and Sherman wetland on the nearby Sherman Island (Fig 1) comprise a meso-scale monitoring network for C, water, and energy flux measurements by eddy covariance [60]. Since the wetlands are below sea level, they lack natural hydrologic connectivity and depend

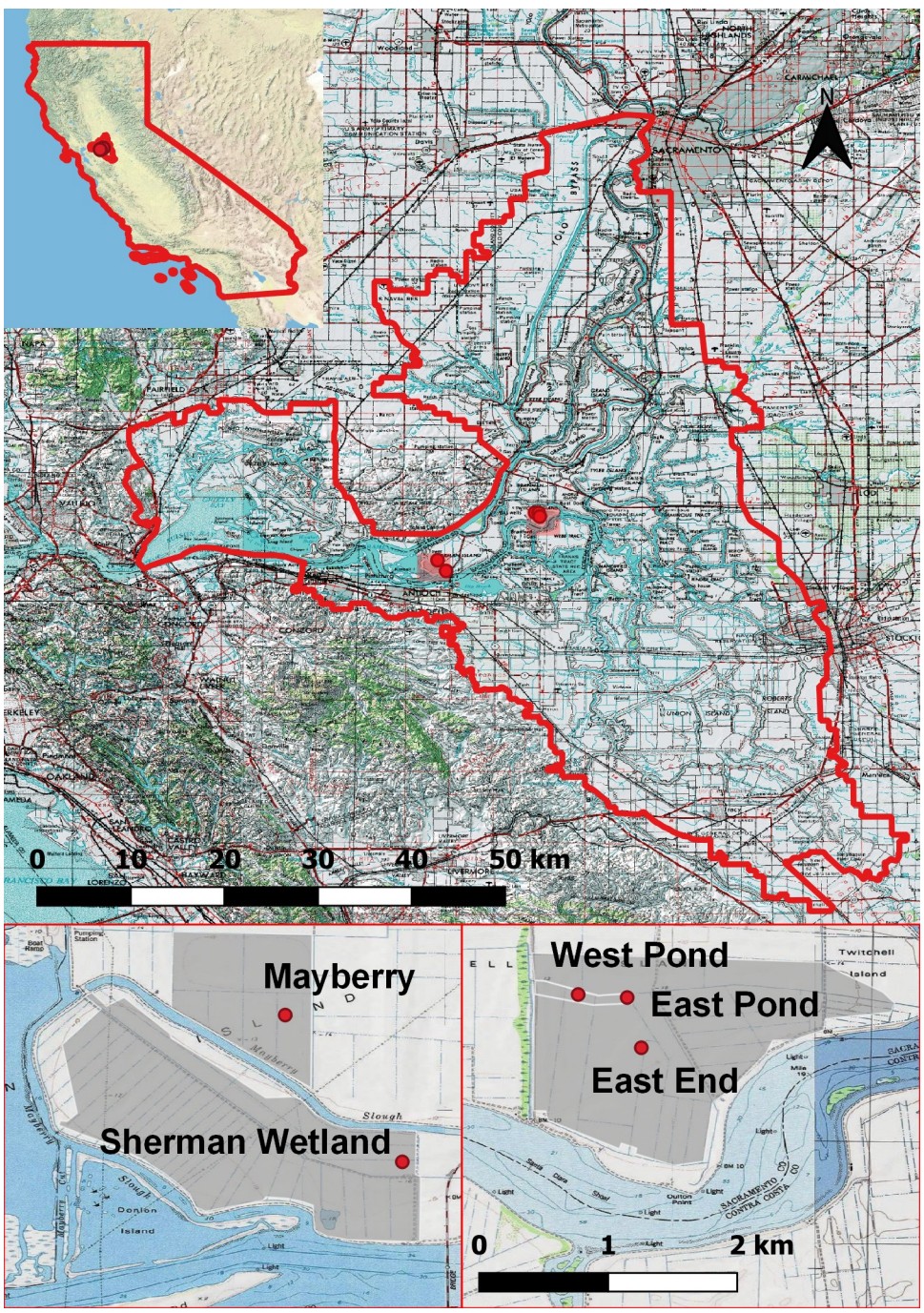

**Fig 1. Map outlining the Sacramento-San Joaquin Delta with the five restored wetland sites.** Site locations are marked (top) and enlarged (bottom) to show the wetland areas (shaded grey) and instrument tower locations (red points).

**Table 1. Site characteristics of all five restored wetlands in the Sacramento-San Joaquin Delta, USA.**

| Site (Ameriflux ID) | Previous land use | Year restored | Area (ha) | Elevation at tower (m.a.s.l) | Water level (cm) | Vegetation establishment method | Plant proportion in 2018: total site (85% flux footprint) | Dominant plant species (%) | Open water areas | Flux data coverage (complete years) |
|---|---|---|---|---|---|---|---|---|---|---|
| West Pond (US-Tw1) | Crops | 1997 | 3 | -4.9 | 25 | Planted sections | 0.997 (0.970) | *Typha*[a] (69%) *S. acutus* (31%) | No | 2013–2018 (6) |
| East Pond (US-Tw5) | Crops | 1997/ 2013[b] | 2.6 | -5.2 | 55 | Planted sections/ leftover rhizomes | 0.968 (0.912) | 59% *Typha*[a] (59%) *S. acutus* (11%) | Yes | 2018 (1) |
| Mayberry (US-Myb) | Pasture | 2010 | 121 | -3.5 | 20, 120 (channels) | Margins planted | 0.644 (0.480) | *Typha*[c] (55%) *S. acutus* (28%) | Yes | 2010–2018 (9) |
| East End (US-Tw4) | Crops | 2013 | 323 | -5.0 | 25, 60 (channels) | Transplanted individuals and seeding | 0.821 (0.842) | *Typha*[c] (95%) *S. acutus* (5%) | Yes | 2014–2018 (5) |
| Sherman Wetland (US-Sne) | Pasture | 2016 | 263 | -4.4 | 10–20, >200 (channels) | Passive wind dispersal with dry areas planted | 0.583 (0.454) | *Typha* (NA) *S. acutus* (NA) | Yes | 2017–2018 (2) |

Variables include previous land use, water table level, vegetation composition, and restoration methods.

*NA*: No vegetation survey data available.

[a]Data taken from a 2008 vegetation survey [64].

[b]East Pond vegetation was completely removed in 2013 to seed East End and is considered to reset the wetland age.

[c]Data from vegetation transects [65].

on a pump system to actively manage water levels and maintain inundation suitable for marsh vegetation. Dominant species include *Schoenoplectus acutus* and various *Typha* spp., of which *Typha angustifolia* is classed as non-native [47,61]. These wetlands have undergone slightly different restoration practices (construction design and vegetation recruitment, see Table 1) and have experienced a range of site conditions and disturbance events. A detailed description of West and East Pond wetlands and more background information on all sites can be found elsewhere [42,44,62]. West Pond and East Pond were restored in 1997 with measurements since 2012 in West Pond and 2018 in East Pond. In 2013, roots and rhizomes were harvested from East Pond and transplanted along the surrounding levees of the neighbouring East End, restored in 2013 from a cornfield. Mayberry was restored in 2010 with a more varied bathymetry including channels (<1.5 m) and open ponds where only the margins were planted. There have been several intermediate-scale disturbance events that have affected the vegetation at this site. In 2014, and to a lesser extent in 2017, there were insect infestations that diminished the green aboveground biomass. During the latter half of 2015 until mid-2017 there was a salinity intrusion and accumulation (daily mean <7 psu), which reduced ecosystem productivity until the wetland was flushed with freshwater [63]. Sherman Wetland was restored in 2016 from a livestock pasture. The bathymetry includes large open water areas and deep channels (>2 m) separating inundated shallow plateaus which were frequently exposed during summer when the water table was lower. Apart from a small area planted with shrubs, the wetland was left to colonise unassisted.

## 2.2 Flux measurements

Continuous eddy covariance flux measurements of CO$_2$ were used to quantify the land-atmosphere exchange of gases [66], along with a set of auxiliary measurements of environmental conditions, including meteorological variables, and soil and water profiles [62]. Fluxes were measured using a suite of sensors at a frequency of 10 (before 2015) or 20 Hz, consisting of

open-path infrared gas analysers (LI-7500 or LI-7500A for $CO_2$ and $H_2O$, LiCOR Inc., Lincoln, NE, USA) that were calibrated every 3–6 months. Sonic anemometers measured sonic temperature and three-dimensional wind speeds at 20 Hz (WindMaster Pro 1352 or 1590, Gill Instruments Ltd, Lymington, Hampshire, England). Quality control, flux processing, and gap filling at these sites has been documented extensively in recent publications [42,63,67].

## 2.3 Flux footprints

Eddy covariance techniques allow for near-continuous spatially integrated measurements of NEE flux responses at high time resolutions. The flux footprint indicates the spatial extent of sources and sinks detected by the flux measurements. The vegetation cover analysis was based on footprints calculated using a basic analytical two-dimensional footprint model [68,69]. This model has been previously validated at these sites and tends to overestimate the extent compared with a more recent model [70]. The area contributing to the measured fluxes fluctuates depending on atmospheric stability conditions and surface parameterisations. Half-hourly footprints were calculated and aggregated to an annual daytime median footprint [71]. Since the flux footprint contributions approach 0 with increasing distance from the tower, only the 85% footprint extent was considered.

Although the sites are ideal for eddy covariance with consistent wind directions, large fetch, and little upwind interference, the flux measurements may not be representative of the whole site because of the high spatial heterogeneity. For example, Mayberry and Sherman Wetland had large open water channels, which were disproportionately represented in the flux footprints compared with the water to vegetation ratio of the whole site. These flux footprints only captured 70–80% of the land cover proportions of the whole site after plant establishment, unlike West Pond, which was very representative with 97–99% similarity.

## 2.4 Vegetation monitoring and classification

**2.4.1 Image classifications.** The land cover classifications within the 85% flux footprint were acquired by Eagle Digital Imaging Inc. and based on images from mid-August of 2014–2018. These orthorectified, mosaicked and radiometrically normalised images covered the red, green, blue, and near-infrared bands at a high spatial resolution (0.15 m). Due to year-to-year variability in site conditions (e.g. wind, water ripples) and vegetation appearance, footprints were classified individually for each year using a semi-automated approach combining automated object-based image segmentation with user-determined decision thresholds and manual classification of objects based on visual recognition in eCognition Developer software v.9 (Trimble Inc.). The Tabulate Area function of ArcMap v10.3.1 [72] was used to calculate the percentage of flux footprints covered by vegetation, open water, and bare soil. In addition, we collected high-resolution aerial imagery (0.05 m) in September 2018 at all sites to distinguish vegetation types within the flux footprints. The impact of image resolution on vegetation indices using these remote sensing methods was recently assessed [73].

The whole site classification to compare the footprint representativeness was done using open access aerial imagery from the National Agriculture Imagery Program (1 m resolution) taken in May or June 2012, 2014, and 2016.

**2.4.2 Canopy height measurements.** The vegetation heights were derived from the aerodynamic canopy height, which was calculated from the turbulence statistics using the methods of our colleagues [74,75]. The analytical equation for aerodynamic canopy height is derived from the logarithmic wind profile based on the Monin-Obukhov similarity theory for a neutral surface layer and assumes parameterisations for the displacement height and surface roughness length [76]. It depends on neutral turbulence conditions and like flux measurements,

results in few valid data points during the winter season, hence we focus on the growing season data. It is susceptible to large outliers at shorter timescales despite careful filtering, so robust averaging was used to avoid bias. Half-hourly measurements were filtered by friction velocity ($0.2 < u^* < 0.5$ m/s), boundary layer stability ($|z/L| < 0.1$ for near-neutral conditions), and time of day (06:00 to 18:00 hours) before being converted to daily and monthly medians. Several methods were used to measure canopy heights and compared in S2 in S1 File. Continuous aerodynamic canopy heights compared the best with available data from several ground-based transect surveys. Ground survey data of plant and litter heights within the flux footprints were only available for West Pond, East End, East Pond, and Mayberry in August 2018 [77]. All methods provide above-water canopy heights.

### 2.5 Statistical analyses

Statistical analyses consisted of Ordinary Least Squares linear regressions and Pearson's correlations with root mean square errors (RMSE) shown for non-linear functions. Parametric ANOVAs with Tukey's Honest Significant Difference tests and non-parametric Kruskal-Wallis tests with Dunn's multiple comparisons (Z-statistics) were used for categorical comparisons (with factor "site") between sites due to the unequal sample sizes [78]. Only full calendar site-years were used in the study and all statistical analyses were conducted using R [79] with packages openair [80], ggplot [81], FSA [82], and dplyr [83]. Uncertainty intervals were estimated by propagating the random half-hourly measurement error and the gap-filling error [42]. The coefficient of variation was used to compare interannual variability of fluxes, whereas a measure of C use or sequestration efficiency, defined as the ratio of annual net to gross ecosystem productivity (NEP/GEP) provided yearly comparisons of net C uptake potentials [84].

## 3 Results

### 3.1 Post-restoration vegetation establishment

The expansion of total vegetation cover within the flux footprint over time since restoration ranged from 0 during initial flooding to a maximum of 0.99 at West Pond (Fig 2), i.e. a mature

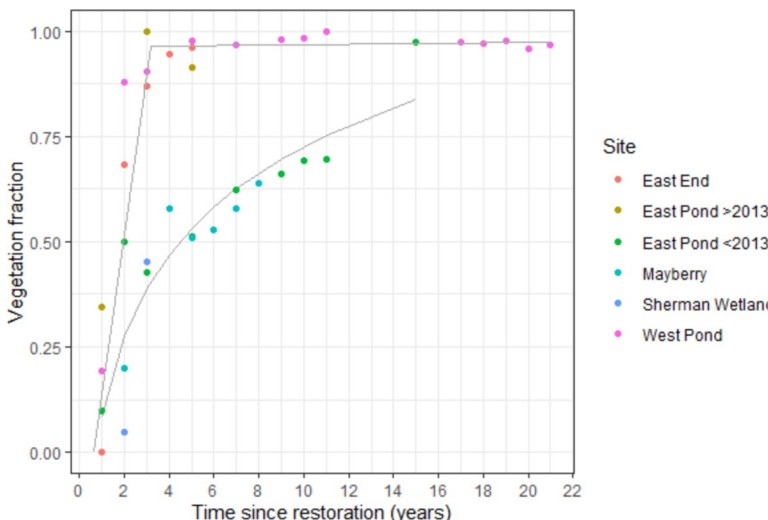

**Fig 2. Proportion of vegetation cover for all wetland sites for each year since restoration.** The trajectories show a two-step linear fit for sites with rapid expansion in the initial establishment phase followed by a post-establishment plateau and a log-linear fit for sites with slow expansion rates. In 2013 East Pond was disturbed to seed the nearby East End wetland and is considered newly restored after this event. These data are based on aerial imagery classifications supplemented by data gathered from ground-based survey transects [44] for West and East Ponds before 2013.

wetland with a dense, closed canopy. Two development trajectories emerged, which were categorised as rapid and slow vegetation establishment responses. Rapid expansion occurred within the first two years after restoration ($R^2 = 0.85$, p <0.05 for the initial linear increase) at West Pond, East End, and East Pond (after the 2013 disturbance) followed by a stable plateau at which the vegetation proportion varied from 0.91 to 0.99 ($R^2 = 0.14$, p = 0.148). For the slow trajectory, vegetation establishment followed a log-linear fit (r = 0.90, p <0.001, RMSE = 0.096) at Mayberry, Sherman Wetland, and East Pond (before 2013), with the cover only reaching 0.7 up to 11 years following restoration.

### 3.2 Land cover classifications

In order to link NEE to the vegetation dynamics, land cover consisting of vegetation, water, and bare soil at each site, as well as vegetation types were classified within the representative flux footprints. There was high spatial variability of land cover and vegetation between site footprints with emergent macrophyte cover types being the most prevalent (Fig 3).

These classifications were compared with monthly and yearly cumulative fluxes corresponding to the time window in which the image used for the classification was recorded (Fig 4). All images were taken during the growing season and corresponding annual NEE showed a linear correlation with net C uptake ($R^2 = 0.46$, p <0.01) within the footprint (Fig 4A), but not a statistically

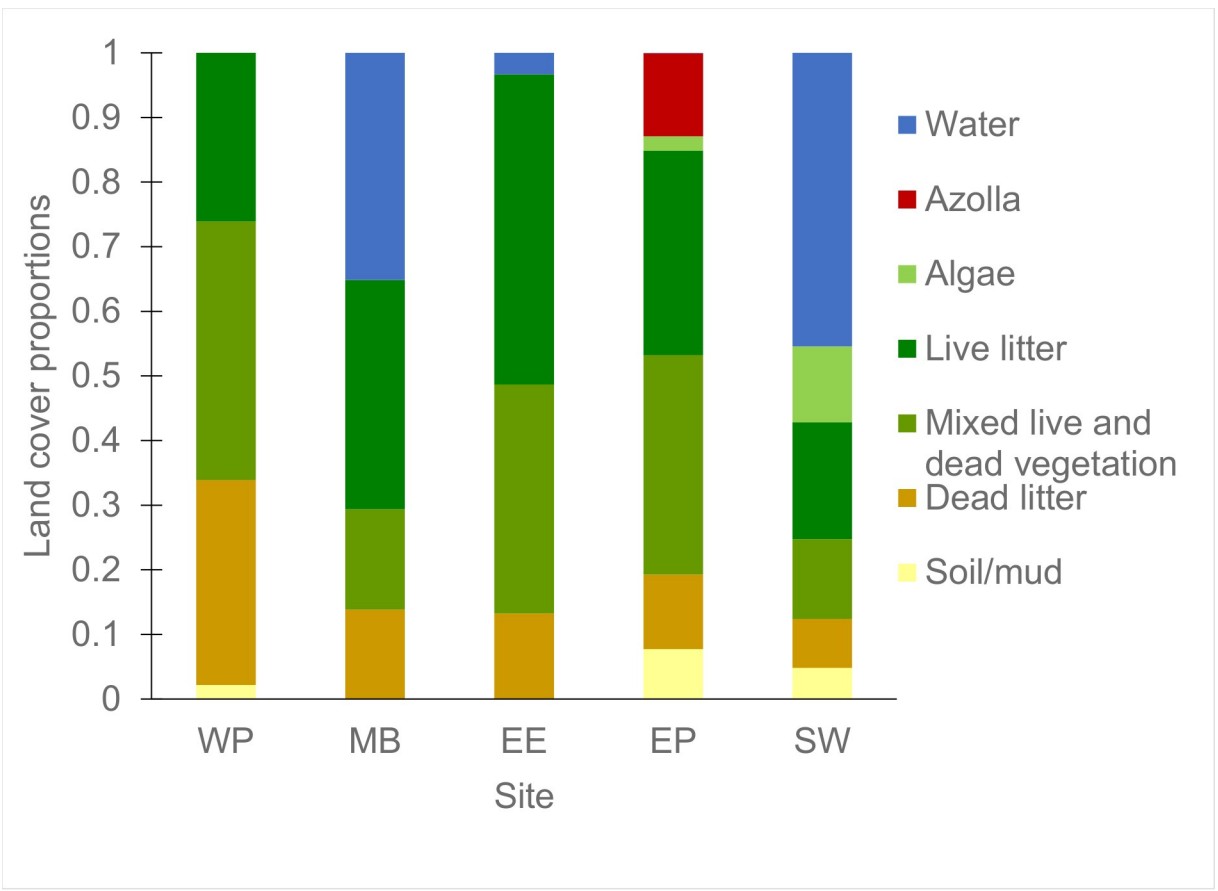

**Fig 3. Land cover proportions split by vegetation type.** This uses September 2018 as an example for all land and vegetation types (live litter, mixed vegetation, and dead litter refer to emergent macrophytes) at all the sites. Site labels are East End (EE), East Pond (EP), Mayberry (MB), Sherman Wetland (SW), and West Pond (WP). See S1 Fig in S1 File for the classified footprints.

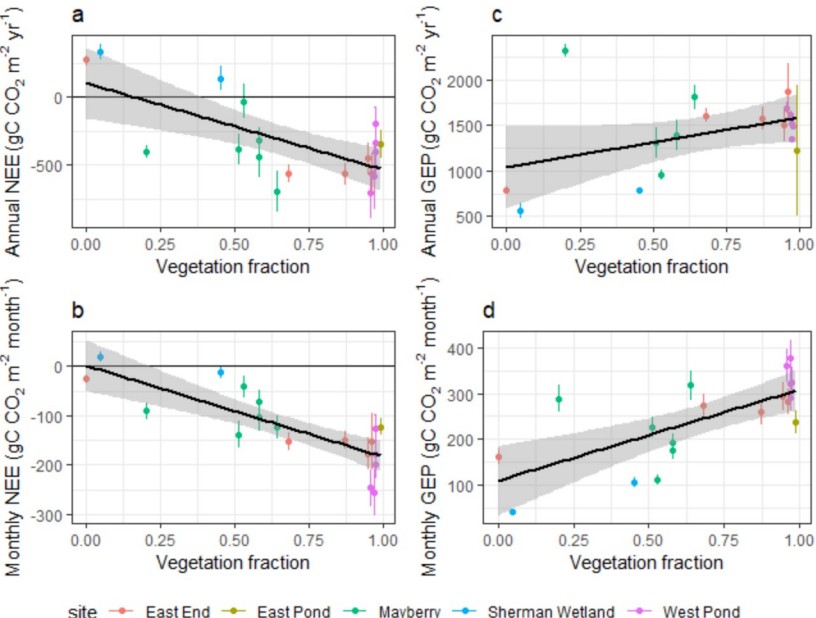

**Fig 4. Flux measurements against vegetation proportions.** Annual (top) and monthly (bottom) cumulative net ecosystem exchange (NEE) of $CO_2$ (left) and gross ecosystem productivity (GEP, right) against vegetation cover during the growing season (May–September) from land cover classifications within the flux footprint for all available site-years with linear regressions and 95% confidence intervals (grey shading for linear regression and bars for data points). Negative NEE values denote carbon uptake by the ecosystem.

significant relationship with annual GEP ($R^2$ = 0.18, p = 0.07, Fig 4C). Vegetation proportion correlated both with monthly cumulative NEE ($R^2$ = 0.65, p <0.001, Fig 4B) and monthly cumulative GEP ($R^2$ = 0.50, p <0.001, Fig 4D). None of the emergent macrophyte litter types (i.e. dead, live, and mixed live and dead biomass) individually showed a statistically significant (p <0.05) or strong ($R^2$ <0.5) relationship with NEE at either timescale. A finer resolution classification showed greater spatial complexity and clear differentiation of vegetation types (S1 Fig in S1 File). Relationships with monthly and annual NEE and GEP during the corresponding timeframe were only marginally improved compared with the lower resolution images of the same period (S1 Table in S1 File), so only the coarser classification results which cover a longer period are discussed. When relating the latest (2018) vegetation cover against the site-mean NEE for established years only, a net C sink was reached with vegetation proportions above 0.55 ± 0.02 ($R^2$ = 0.86, p <0.001, based on data shown in Fig 3 and site-means calculated from annual means in Fig 4A).

## 3.3 Canopy heights

West Pond had the tallest aerodynamic canopy height with a median (interquartile range, IQR) of 3.05 (2.94–3.28) m, while Sherman Wetland had the shortest canopy of 0.18 (0.16–0.26) m. Median aerodynamic canopy height differences were statistically significant for all sites (absolute Z-statistics between 2.73 to 14.53, p <0.01) with East Pond and Mayberry being the most similar (median (IQR) of 2.22 (2.09–2.40) m and 1.85 (1.75–1.96) m, respectively) followed by East End and West Pond (2.77 (2.66–2.91) m and 3.05 (2.94–3.28) m, respectively)).

When plotted as monthly medians for the whole measurement periods (Fig 5), aerodynamic canopy heights clearly displayed canopy growth each year and, where available, plant recruitment after restoration (e.g. East End in 2014, Mayberry in 2011, and Sherman Wetland in 2018). Like vegetation extent, growing season median aerodynamic canopy heights

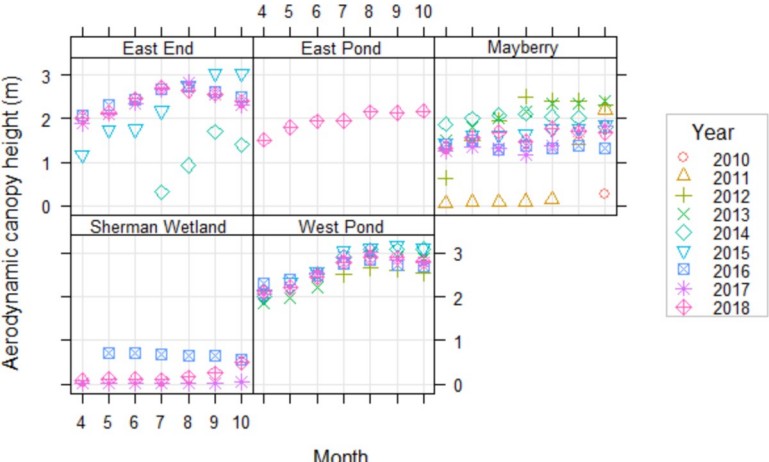

**Fig 5. Monthly median aerodynamic canopy heights.** Canopy heights (m) were calculated using turbulence statistics from the eddy covariance towers for all site-years during the growing season from April to October. Eddy covariance measurements were available for Sherman in 2016 (blue square) before it was converted to a wetland and showed the canopy height of the previous pasture.

explained around half of annual NEE ($R^2$ = 0.53, p <0.001, based on data from Fig 4A and growing season means from Fig 5).

### 3.4 Interannual carbon flux variability

Annual cumulative NEE fluxes showed large interannual variability at some of the sites, such as Mayberry and West Pond (Fig 6). Mayberry had more frequent variations between years (coefficient of variation, CV, of 0.86, n = 8), while the high CV at East End (CV of 0.98, n = 5) reduced to 0.1 when excluding the restoration year, thereby becoming the most stable C sink

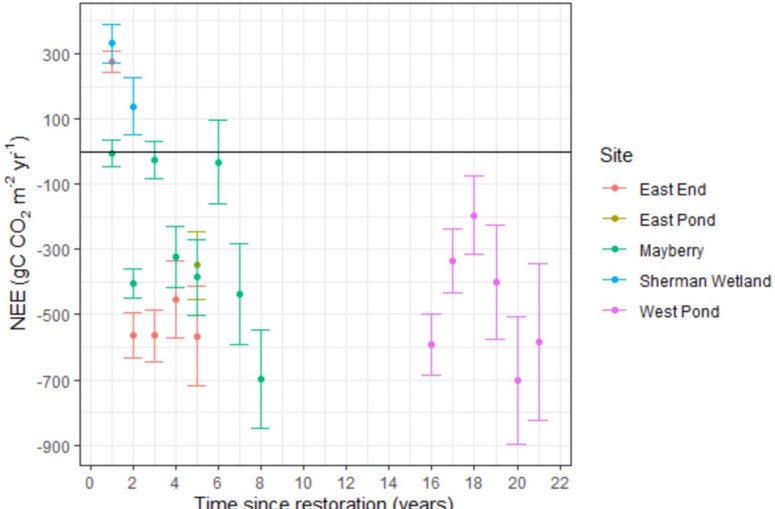

**Fig 6. Annual sums of net ecosystem exchange (NEE) against wetland age (years since restoration).** All complete site-years are shown with bars indicating 95% confidence intervals from propagated errors of the flux measurement and gap-filling methods. The sums for the restoration year (year 0) are not shown (i.e. incomplete years). Negative NEE values denote C uptake by the ecosystem.

with a site-mean (± SD) annual NEE of -536.4 ± 55.6 gC $CO_2$ m$^{-2}$ yr$^{-1}$. West Pond remained a steady sink of -467.8 ± 189.8 gC $CO_2$ m$^{-2}$ yr$^{-1}$ (CV = 0.41, n = 6), whereas Sherman Wetland was a C source of 234.7 ± 137.4 gC $CO_2$ m$^{-2}$ yr$^{-1}$ (CV = 0.41, n = 2).

A measure of C use or sequestration efficiency (i.e. the ratio of net to gross ecosystem productivity, NEP/GEP) ranged from -59% to 41% (negative values indicate C loss) with site means ± SD (and mean ± SD excluding the initial establishment year) of -38 ± 29 (-17) % (n = 2 and 1) for Sherman Wetland, 18 ± 15 (21 ±17) % (n = 8 and 7) for Mayberry, 19 ± 30 (33 ±3) % (n = 5 and 4) for East End, 29 ± 10% (n = 6) for West Pond, and 31% (n = 1) for East Pond. The positive linear regression of GEP and NEP with all site-years was moderately strong ($R^2$ = 0.65, p <0.001, slope = 0.58) and improved to $R^2$ = 0.89 (p <0.001, slope = 0.78) when the two outlying points (Mayberry 2012 and 2013) were excluded.

## 4 Discussion

Net cumulative C fluxes on annual time scales, are a key metric for comparing and evaluating restored ecosystems for their C sequestration capacity, as it shows the net effect of multiple processes representative at the ecosystem-level and highlights the differences in their functioning. All wetlands were dominated by emergent macrophytes, but differed in their distribution, rates of expansion, as well as extents of other vegetation types, which we used to link variations in C fluxes between sites and years.

### 4.1 Restoration design impacts initial vegetation expansion rates

The trajectories of vegetation establishment after restoration could be explained by the differences in restoration methods, i.e. different water levels from undulating bathymetry design and planting patterns. Our colleagues [44] showed that relatively small changes in water depth (25 vs. 55 cm) affected plant productivity at East and West Pond with faster colonization of shallow areas by emergent marsh vegetation. Another study [85] also indicated that there are different hydrologic constraints between *Schoenoplectus* and *Typha* species with both water levels and flooding duration affecting survival rates. Sites with shallow root zone water levels (West Pond, East Pond post-2013, and East End) followed the rapid linear establishment trajectory reaching a constant extent around 95% within 2 years. Although all three sites contained transplanted patches promoting fast colonization, only after the 2013 disturbance did East Pond follow the rapid trajectory. This was likely from the faster regrowth of leftover rhizomes, since asexual clonal expansion is more effective than recruitment from seedbanks [86], whereas before 2013 the deeper water level hindered plant recruitment. Sites with deeper water (East Pond pre-2013) and/or larger open channels (Mayberry and Sherman Wetland) followed the slower rate of expansion. It is important to note that flux datasets generally only provide the water level measured by the flux towers, which can be a poor indicator of site-wide water depth for heterogeneous sites with large variations in bathymetry. With bathymetry design and water depth being key factors of vegetation recruitment after restoration, it is vital to reduce bathymetry variations to achieve higher vegetation to water ratios when designing wetlands for C sequestration, as well as collect elevation data or more spatially resolved water depths to accurately evaluate C sequestration capabilities.

Nutrient availability may also have influenced establishment rates. Based on soil samples, we found that East End had very high nutrient levels, while Mayberry or Sherman Wetland indicated low levels (mean ± SD C:N ratios of 16.7±1.4, 15.7±0.8, and 13.1±0.2, as well as NaOH-extractable inorganic phosphorus (P) of 599 ± 105.8 compared to 152.2 ± 62.2 and 298 ± 82.6 μg P g$^{-1}$ dry soil, respectively; S3 in S1 File). Although East End also had undulating bathymetry and only the levees were transplanted with *S. acutus*, *Typha* spp. spread rapidly by

outcompeting other species under high P conditions because they are hyper-accumulators [87,88]. High P conditions (>500 μg g$^{-1}$) have been shown to result in highly productive, but low diversity plant communities, which benefits C uptake, but counteract biodiversity targets of restoration [4,89].

The vegetation expansion functions agreed with the second hypothetical trajectory for a post-restoration vegetation index [90] and highlighted that colonisation is often non-linear. Unless rapid initial vegetation establishment can be accomplished within the first years, the final equilibrium state can take >20 years to fully unfold [91] and may require longer monitoring to assess ecosystem service capabilities [65]. Even then, restored wetlands may never reach pre-disturbance states and levels of functioning [92].

## 4.2 The contribution of smaller vegetation types and areas with dead litter to net carbon uptake is non-negligible

The wetland designs caused significant spatial heterogeneity between levels of plant and litter types, even though emergent macrophytes were dominant at all sites (Fig 3). West Pond had the highest relative proportion of dead litter and the lowest live vegetation reflecting both the age of the wetland and the lack of open water areas leading to a dense closed canopy with less visibly green biomass. Despite the significant litter build-up reducing and/or delaying GEP, it remained a strong C sink. Across all wetlands both cumulative monthly and annual mean NEE fluxes showed the best correlations with the total vegetation cover which included dead litter and smaller plant types. Dead and mixed litter may have obscured new underlying growth that was contributing to overall GEP [90,93]. Several studies [94,95] found that erectophyle plants, such as *Typha* and *S. acutus* had higher reflectance in visible wavelengths as the tips of the plants brown first which can conceal underlying photosynthetically active biomass in aerial images limiting the effectiveness of remotely-sensed C flux estimates in wetlands [96].

Although emergent macrophytes are dominant primary producers [97,98], other species, such as floating aquatic vegetation (e.g. *Azolla* and *Lemna*) and algae accounted for 16% and 24% of the total vegetation cover at East Pond and Sherman Wetland, respectively. Their inclusion improved the relationship between plant cover and NEE ($R^2$ from 0.48 to 0.65) indicating that these species provided a non-negligible contribution to overall NEE. Although small-statured, such aquatic species can sequester considerable amounts of C. For example, *Azolla*, when cultivated as a dual crop with rice, was found to sequester additional 168 g $CO_2$ m$^{-2}$ yr$^{-1}$ [99] and may have been an important C sink at global scales during the Eocene [100]. Among other benefits, it has been shown to reduce other GHG emissions, such as $CH_4$, $N_2O$, and water vapour from rice crops [101,102]. *Lemna*, which is seasonally widespread in the Delta wetlands, can also promote methanotrophy [103], as well as sequester around 3.5 gC $CO_2$ m$^{-2}$ d$^{-1}$ during the peak growing season [104]. Many algal species have proved to exhibit a large capacity for $CO_2$ sequestration [105,106] but can deplete the dissolved oxygen levels in the water leading to increased $CH_4$ emissions and fish mortality. Floating aquatic vegetation cover can change rapidly during the growing season [107], therefore the snapshot aerial imagery used may not reflect their complete extent and contribution to the annual budgets. Due to the fast growth rate and extensive coverage it may be beneficial to promote *Azolla* and *Lemna* in restored wetlands to boost C uptake from otherwise unproductive open water areas [108].

## 4.3 Dense canopies may dampen carbon uptake variability

Monthly median aerodynamic canopy heights for all site-years clearly showed canopy growth during the initial vegetation establishment after restoration, e.g. East End in 2014, Mayberry in 2011, and Sherman Wetland in 2018, as well as boom years immediately following the initial

growth year, i.e. East End in 2015, Mayberry in 2012 and 2013 (Fig 5). Most canopies increased by 1 m during the growing season peaking in August before experiencing die-back and wind-fall in September and October with open water sites having overall shorter median canopy heights (<2 m) and closed canopies being taller (3 m). Although closed canopy sites had more dead and mixed litter patches, the higher canopy heights likely compensated any reduction in GEP. Some years also indicated a mid-season dip, such as Mayberry 2016 and 2017 likely due to disturbance events that limited plant growth, i.e. the salinisation event in 2016 [63] and an insect infestation in 2017.

The minimum canopy height at the start of the growing season represents the persistent lit-ter layer that exists year-round and required around 2 years to accumulate. Taller canopies also translated to a higher dead litter layer. Our colleagues [77] showed that the plant area index (PAI), a proxy for leaf area index, at West Pond and Mayberry reflected litter height, which remained constantly high from the ground until the top of the litter layer showing how uniform and very dense the litter is throughout the canopy. Above this layer the PAI rapidly declines indicating the openness of canopies with erectophyle species. Due to emergent macro-phytes remaining erect after senescence, the resulting litter build-up may introduce greater complexity and asynchrony between the relationships of NEE with environmental drivers. It is possible that this dense layer, taking years to build up, can insulate the below-canopy environ-ment, such as by attenuating light, creating a stronger temperature gradient within the canopy, and retaining humidity. By altering the biophysical properties, the layer can impact gas exchange [109], such as reducing respiration rates which would also explain the higher NEE despite dead vegetation patches. In their supplemental information, a previous study [110] showed that West Pond, the oldest wetland, had more asynchronous NEE driver responses at longer multiday to seasonal timescales compared to Mayberry, which was a young wetland at the time. Wetlands with more dense canopies were also more consistent C sinks regardless of age. This disconnect may lessen the effects of weather extremes on C uptake but increase responses to local ground conditions possibly explaining some of the large differences between sites and years when meteorological conditions were comparable.

## 4.4 Productive marshes are large net carbon sinks, but disturbance can strongly reduce sink strength

Site-mean annual NEE balances and radiative forcings were discussed and compared else-where [42,14] and showed that these wetlands can be GHG sinks when net C uptake is high enough to offset CH$_4$ emissions. Hence, this study focusses on how the C uptake variability between individual years was related to vegetation dynamics which can be managed to maxi-mise the wetland's climate change mitigation function. It is important to note that GEP is not an independent measure, as it is calculated by the difference of partitioned ecosystem respira-tion from night time NEE fluxes [111], and may be overestimated by around 15% during the growing season using this partitioning method [112].

Most sites became a net C sink two years after restoration under both vegetation establish-ment trajectories with a total vegetation cover of 0.50 for the rapid and 0.28 for the slow trajec-tories, which corresponded to a cross-site mean NEE for the second years of -562 ± 35.5 gC m$^{-2}$ yr$^{-1}$ for the rapid and -133 ± 383 gC m$^{-2}$ yr$^{-1}$ for the slow trajectory wetlands, respectively. The sites with rapid vegetation expansion (West Pond, East End, and East Pond after 2013) were all consistent C sinks, whereas the slower expanding systems had C source years (Sher-man Wetland, where vegetation had still not fully established after 2 years) or had greater interannual variability with some years being near C neutral (CV of 0.86 for Mayberry), most of which could be attributed to disturbance events. Net C uptake required on average 55%

vegetation cover, which both trajectories achieved (at 2 and 5 years for the rapid and slow expansion trajectories respectively). Overall, vegetation proportion only accounted for less than half of year-to-year variability in NEE once plants were established. NEE only scaled with vegetation extent in the initial years, after which other factors became more dominant drivers representing a succession of NEE flux drivers. If restoration plans design a bathymetry allowing for a minimum of 55% plant cover, although potentially not reducing net C uptake, the channels and ponds will likely not accrete soil as fast as the surrounding vegetation creating larger depth gradients between the water and plants. This may lead to future maintenance issues, such as increased water use to sustain sufficient inundation for peak plant productivity and avoid drought stress.

Considerable interannual variability in C sink strength ranging from -718 ± 72.4 to -25.9 ± 55.5 gC $CO_2$ $m^{-2}$ $yr^{-1}$ after vegetation establishment was seen at all sites, even 20 years post-restoration, as was found in other restored wetlands ranging from -481 to 572 gC $CO_2$ $m^{-2}$ $yr^{-1}$ [113]. This agrees with the range of net C balances from undisturbed freshwater marshes of -978 to 43 gC $m^{-2}$ $yr^{-1}$ with a median around -272.1 gC $m^{-2}$ $yr^{-1}$ [26]. For most of the low uptake years in the Delta, disturbance events, including salinization [63], low water levels, and insect infestations, were responsible for lowering GEP. In other years, carryover effects were found to impact C uptake causing low C sequestration efficiencies (NEP/GEP ratio), which is determined by the balance of GEP and $R_{eco}$. Such a dynamic was found at Mayberry, i.e. years 2 (2012) and 3 (2013), which had no clear indications of disturbance, but resulted in a net C neutral year in 2013. Mayberry experienced the largest vegetation growth and highest GEP in 2012, which was accompanied by higher than average $R_{eco}$ relative to GEP resulting in moderate net C uptake. In 2013, however, high $R_{eco}$ matched the high GEP leading to a C neutral year. These early years show a boom and bust cycle (Fig 6), wherein both autotrophic and heterotrophic respiration was promoted initially via the expansion of biomass including belowground roots and the input of fresh root exudates driving soil microbial activity. The following year, these easily-degradable C inputs from the excessive build-up of dead material from the previous year provided additional substrates to elevate $R_{eco}$ compared to GEP [114]. Another study [28] also saw successional patterns in aboveground biomass in restored wetlands with planted vegetation, where the unplanted wetland was more productive 10 years later but had lower diversity and was more susceptible to stress. Because soil C in wetlands is mostly protected by anaerobic conditions, disturbances, especially involving changes in water levels, can cause disproportionately large C losses both from increasing $R_{eco}$ and reducing GEP, which has been observed in other disturbed wetlands [115,116 and references therein]. Since wetland C is more susceptibile than non-saturated systems, it is crucial, where possible, to manage wetlands to mitigate the impacts of disturbance or extreme events on GEP to maintain high C sequestration, as well as minimise $R_{eco}$. Management activities depend on the wetland design, but examples may include maintaining water levels to fully inundate soils, while protecting plants from salt intrusions and accumulation in freshwater systems, pest and pathogen infestations, etc. [14].

Generally, the C sequestration efficiency increased as wetlands aged. In several post-establishment site-years the C sequestration efficiency approached the theoretically constrained value of 0.3 (i.e. half the maximum value) and was similar to those measured in other aquatic systems [117] and nutrient-rich upland forest ecosystems [118]. The overall slope of NEP to GEP (slope = 0.58) approached that found in a synthesis of 92 forest sites (slope = 0.73), which are considered to have the highest climate mitigation potential in terms of C sequestration [22]. Considering the substantially longer residence time of C sequestered in wetlands (>1000 years) compared to upland forests (~100s of years), our results highlight that productive marshes provide important contributions towards C sequestration goals, especially over

climate-relevant timescales. Although wetland restoration creates additional greenhouse gas emissions, these can either be reduced by management strategies (e.g. Table 2 in reference 14) or be offset partially or completely if productivity is high enough to reach an immediate net GHG sink status, as well as a future GHG sink using cumulative GHG emission metrics. Several site-years were shown to already be small to moderate net GHG sinks even with high $CH_4$ emissions, due to the high NEP only a few years after restoration [14].

## 5 Conclusions

Wetland restoration as a negative emission technology requires systems to remain net C sinks over decadal to century timeframes. There is an urgent need to quantify and understand ecosystem-scale C fluxes from complex heterogeneous wetlands over longer timeframes, as studies have shown wetlands to vary from being strong C sinks to considerable C sources. To better understand the drivers of variability in C sink strength related to restoration design and vegetation dynamics, we investigated eddy covariance $CO_2$ flux measurements covering 21 site-years of five highly productive restored marshes in a Mediterranean climate that represent a range of post-restoration stages. Our results showed that established wetlands were strong net C sinks even with high interannual variability in NEE decades after restoration.

Overall, we found that restored wetlands generally became strong C sinks as early as two years after restoration and with a vegetation cover >55%. The trajectory of vegetation establishment depended on restoration design, such as bathymetry-determined water levels and planting methods with possible legacy effects, such as nutrient inputs, from previous land use. Furthermore, seemingly unproductive patches of dead litter or small floating aquatic vegetation and algae provided a non-negligible contribution to annual net C uptake. However, the relationship of plant cover to NEE only explained around half of the interannual variability, with a stronger relationship during establishment years, indicating a succession of dominant NEE flux drivers. Canopy height, as estimated from turbulence statistics, provided insight on vegetation structure, which increased in complexity with wetland age and showed boom-bust cycles that explained undisturbed years with low C uptake. As wetlands matured, the dead litter layer created a buffer between the atmosphere and the below canopy environment possibly causing lag effects between environmental drivers of NEE which may strengthen the C sink stability. This may explain one process by which small-scale effects became more dominant drivers of NEE and some of the large variability between years when meteorological conditions were similar.

Our data showed that restored wetlands within the same climate varied significantly in their functioning, including net C uptake rates and flux drivers, which depended on the restoration design, wetland age, and disturbance events. Overall, the net C uptake capabilities of these wetlands were large but, especially wetlands with lower vegetation cover and hence C uptake, were susceptible to local disturbance events. We stress the importance and need of such site-specific characteristics to be considered in wetland restoration designs and post-restoration management. The high C sequestration efficiency, on average 20–30% of GEP, was comparable to that of upland systems, indicating that such productive marshes if well-managed may be more efficient for C sequestration than upland systems, as well as provide a wide extent of co-benefits even from small wetlands areas.

## Supporting information

**S1 File. Contains supporting text, figures, and tables.**
(DOCX)

## Acknowledgments

We thank the editor and reviewers for their insightful and constructive feedback which greatly helped to improve this manuscript. The authors recognise the work of all past and present Berkeley Biometeorology Lab members who helped maintain towers and collected and processed data over the lifetime of these sites, as well as the undergraduate summer lab assistants. We thank the Metropolitan Water District of Southern California for collaboration and access to the research sites.

## Author Contributions

**Conceptualization:** Alex C. Valach, Kuno Kasak, Kyle S. Hemes, Dennis D. Baldocchi.

**Data curation:** Alex C. Valach, Kuno Kasak, Kyle S. Hemes, Iryna Dronova, Sophie Taddeo, Daphne Szutu, Joseph Verfaillie.

**Formal analysis:** Alex C. Valach, Kuno Kasak, Iryna Dronova, Sophie Taddeo.

**Funding acquisition:** Whendee L. Silver, Dennis D. Baldocchi.

**Investigation:** Alex C. Valach, Kuno Kasak, Kyle S. Hemes.

**Methodology:** Alex C. Valach, Kuno Kasak, Kyle S. Hemes, Tyler L. Anthony, Iryna Dronova, Sophie Taddeo.

**Project administration:** Alex C. Valach, Dennis D. Baldocchi.

**Resources:** Alex C. Valach, Whendee L. Silver, Daphne Szutu, Joseph Verfaillie, Dennis D. Baldocchi.

**Software:** Alex C. Valach.

**Supervision:** Whendee L. Silver, Dennis D. Baldocchi.

**Validation:** Alex C. Valach.

**Visualization:** Alex C. Valach, Iryna Dronova, Sophie Taddeo.

**Writing – original draft:** Alex C. Valach, Kuno Kasak.

**Writing – review & editing:** Alex C. Valach, Kuno Kasak, Kyle S. Hemes, Tyler L. Anthony, Iryna Dronova, Sophie Taddeo, Whendee L. Silver, Daphne Szutu, Joseph Verfaillie, Dennis D. Baldocchi.

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
