## [Decision Letter · Decision Letter 0]

17 Jan 2021

PONE-D-20-35610

Wetland restoration for natural climate solutions: Productive wetlands quickly become net CO2 sinks, but multiyear sink strength and stability vary with site-level factors related to restoration design and management

PLOS ONE

Dear Dr. Valach,

Thank you for submitting your manuscript to PLOS ONE. After careful consideration, we feel that it has merit but does not fully meet PLOS ONE’s publication criteria as it currently stands. Therefore, we invite you to submit a revised version of the manuscript that addresses the points raised during the review process.

We look forward to receiving your revised manuscript.

Kind regards,

Hojeong Kang

Academic Editor

PLOS ONE

Additional Editor Comments:

The reviewers were generally positive about the paper, but raised several issues that should be addressed. I'd recommend the authors to address them thoroughly and revised the manuscript accordingly.

Journal Requirements:

"NO"

4. We note that Figures 1 and Supplementary Figures S1-3 in your submission contain map images which may be copyrighted. All PLOS content is published under the Creative Commons Attribution License (CC BY 4.0), which means that the manuscript, images, and Supporting Information files will be freely available online, and any third party is permitted to access, download, copy, distribute, and use these materials in any way, even commercially, with proper attribution. For these reasons, we cannot publish previously copyrighted maps or satellite images created using proprietary data, such as Google software (Google Maps, Street View, and Earth). For more information, see our copyright guidelines: http://journals.plos.org/plosone/s/licenses-and-copyright.

4.1.    You may seek permission from the original copyright holder of Figures 1 and Supplementary Figures S1-3 to publish the content specifically under the CC BY 4.0 license. 

4.2.    If you are unable to obtain permission from the original copyright holder to publish these figures under the CC BY 4.0 license or if the copyright holder’s requirements are incompatible with the CC BY 4.0 license, please either i) remove the figure or ii) supply a replacement figure that complies with the CC BY 4.0 license. Please check copyright information on all replacement figures and update the figure caption with source information. If applicable, please specify in the figure caption text when a figure is similar but not identical to the original image and is therefore for illustrative purposes only.

Reviewers' comments:

Reviewer's Responses to Questions

**Comments to the Author**

1. Is the manuscript technically sound, and do the data support the conclusions?

Reviewer #1: Yes

Reviewer #2: Yes

Reviewer #3: Partly

2. Has the statistical analysis been performed appropriately and rigorously? 

Reviewer #1: Yes

Reviewer #2: Yes

Reviewer #3: No

3. Have the authors made all data underlying the findings in their manuscript fully available?

Reviewer #1: Yes

Reviewer #2: Yes

Reviewer #3: Yes

4. Is the manuscript presented in an intelligible fashion and written in standard English?

Reviewer #1: Yes

Reviewer #2: Yes

Reviewer #3: Yes

5. Review Comments to the Author

Reviewer #1: This study combines eddy covariance data from five different restored peat-rich wetlands in the San Joaquin delta to evaluate the carbon sink potential of restored wetland as a climate change mitigation option. The authors also compile data on plant cover within the flux footprints to better understand simple indicators of when restored sites will act as carbon sinks. The authors found that wetlands quickly became carbon sinks, but the sink strength varied between years, which they refer to as “fragility”. Overall, the results are sounds and this is a valuable data set used to interrogate management options. Aside from some minor corrections, I suggest that the authors also more clearly demonstrate how fragile the carbon sink function of these restored sites is. They compare these systems to forests, but it would also be worthwhile to place the interannual variability into the context of the interannual variability that would be observed in an undisturbed (or at least less disturbed) marsh system. I also think that making some clearer management recommendations/policy recommendations at the end of the paper could strengthen it overall (i.e., should marsh restoration be considered a climate change mitigation strategy? What does “well-managed” look like/require?).

Specific comments:

Line 84: How much does the past management actions affect the post-restoration outcome (i.e., could the same restoration approaches result in different outcomes depending on the management during agricultural use)?

Line 90-91: Is this really a fair assumption? At least in freshwater inland peatlands these losses can be a very important component of the C budget (Roulet et al 2007, Evans et al 2016).

Line 98: But will those values be included here again for management considerations?

Line 271: I think it should be “NEE and GEP during the corresponding…”

Lines 320-321: Isn’t the correlation between GEP and NEP confounded since the former is calculated from the latter? Do you really need this correlation to explain the findings?

Line 358: I suggest rewording this as something like “Based on soil samples, we found that East End had very high nutrients…”

Line 377: What “despite”. I would guess that the litter build-up is a good indicator that the site is a C sink (i.e., you are visually seeing the C accumulate in the accumulating litter). I guess, the thought is that the litter is a new source for C release via respiration?

Line 437: I’m not sure you really show the fragility in this section. It’s true that they were C sources in some years, but this was mainly in the early years post-restoration or during known disturbances. There are also risks for other types of C sinks (like forests mentioned in this section), so I think the reasons marshes might be more fragile needs to be more clearly explained in this section, or how you define this interannual variability needs to be reworded.

Reviewer #2: Summary. Freshwater peatlands store large amounts of carbon in their soils and vegetation, and there is growing interest in using wetland restoration, conservation and management techniques as a climate mitigation strategies. This paper takes advantage of long-term eddy covariance CO2 flux measurements and associated measurements of vegetation community and restoration design to measure CO2 sink strength in 5 wetlands. All 5 wetlands experience similar climate conditions, allowing the authors to explore site-specific controls of CO2 sink. The manuscript shows that vegetation recovery followed 2 distinct trajectories based on restoration design and that most became a CO2 sink after reaching 55% vegetation cover. The manuscript offers additional insights into the temporal variability of sink strength. The manuscript is timely and generally well written. I offer a few comments below.

General Comments.

The title is a mouthful. I don’t think it is ‘wrong’ per se, it is just really long. I have to wonder if there is any way to trim it down and still give the reader a sense of the story.

The authors are clear that in the current MS they are focusing on time-series of CO2 fluxes to explore their controls and that they are not attempting to conduct a fully greenhouse gas accounting for these systems (e.g., L96-98; L438-440). This decision makes sense to me, especially considering that radiative forcing has been explored elsewhere, but it does limit the ability of the authors to make a strong case for climate mitigation in these systems and explore their use as ‘natural climate solutions’. Is there is a chance to fold in a bit more about CH4 and N2O fluxes in the discussion to further make the case that there is, indeed, a net climate benefit for restoration of these systems once they become net CO2 sinks? A full greenhouse gas balance is beyond the scope of the paper, but perhaps some broad brushstrokes for context?

I found the paragraph beginning in L389 to come out of nowhere. This is the first mention of Azolla in the entire manuscript aside from in the legend of Figure 3 (where it looks to cover ~15% in a single site). The authors then give a fair amount of space to Azolla, including conversation around its impacts on CH4 and N2O fluxes, which are not even measured in the current work. I take the point that floating plants and algae are non-negligible contributors to NEE in the study, but this discussion felt really disjointed from the rest of the paper to me.

Minor comments:

L17. Consider replacing “litter decomposition rates” with “decomposition”. It isn’t just litter that decays slowly (so does SOM, if litter and SOM are actually different things in a peatland?) and I think the word rate is redundant in this context.

L21. Consider removing “potentials”. You show that restoration impacts C sequestration, not just the potential for this process, right?

L27. Consider removing “and rates”

L31. Can you clarify what you mean by “which most wetlands achieved with vegetation establishment”? Are you saying that most systems become sinks once they hit the 55% threshold?

L32. I might add in the equation that you used to define sequestration efficiency (NEP/GPP) here.

L42-44. Estimates of carbon stocks in northern peatland soils have recently been expanded. Might be worth looking at -- Nichols, J.E., Peteet, D.M., 2019. Rapid expansion of northern peatlands and doubled estimate of carbon storage. Nature Geoscience 12, 917–921.

L50. I think you’re missing a word here. Perhaps “…uncertainty ABOUT whether….

L62. Can you replace the subject, “This”, with something more concrete? Something like “Exploring these site-specific controls requires….”

L65. I think you can rewrite this without making the authors (“us”) a part of the story. “The eddy covariance method allows for direct measurements of net ecosystem exchange (NEE) over an integrated ecosystem-scale area, and generally assumes that the sources are homogeneous and representative of the whole site.”

L81. Consider removing configurations.

L112-113. I think you need to add a comma after irreversible

L138-139. I think you need to add a comma after 2014. I also think that since there were multiple outbreaks you need to use were here. “In 2014, and to a lesser extent in 2017, there WERE insect infestationS that diminished the green aboveground biomsass.”

Table 1. Can you clarify why the dominant plant species in the Sherman Wetland is NA in this table? There is clearly some vegetation there (58% coverage). What does NA mean here?

L156. I would not discuss the eddy covariance methods used to measure CH4 if those data are not going to be considered in this paper.

L177-178. This language is redundant

L263. Can you clarify what is meant by “with vegetation cover >0.1” here? When I look at Figure 4a, it looks like your linear relationship was built with all of the data, including sites with lower vegetation cover. What am I missing here?

Figure 4. Just a note that you discuss this figure in the order 4a, 4c, 4b, 4d in the text. Do you want to move the panels to align with this order? I am also a bit confused by the units used on the vertical axis in Figure 4b and 4d. In the text, you suggest that you are looking at controls of monthly CUMULATIVE NEE and GEP. Yet, the axis units are PER MONTH. If this is a cumulative sum of multiple months, how can it be a per month number?

L167 (and elsewhere). My understanding is that i.e. (and e.g.) should be followed by commas “i.e.,” as these are abbreviations for phrases.

L269-273. I think this is a run-on sentence. Consider breaking it up and starting a second sentence with “Hence, we only discuss…”

L358-364. I appreciate the decision to put the detailed nutrient information in the supplement to streamline the story in the main body of the manuscript. However, I wonder if there is a way to be a bit more specific about nutrient levels in the main text. Could you provide summary statistics instead of “very high” and “low” when discussing nutrient levels.

L396. Replace “methane” with “CH4” – you’ve already used the abbreviation elsewhere

L484. Remove “here”.

L489. Replace “methane” with “CH4”

Reviewer #3: This is a fascinating paper, and I have no comments on the details. The manuscript is well written; the analysis is clear, the methods sound and the conclusions are based on the evidence provided.

However, I have one concern about the manuscript utility. If the goal is to assess the vegetation dynamics through time and how it affects the CO2 sequestration that is fine, this is of little use for wetlands and is only part of the story. This is especially true as the authors' selling point is the importance of knowing how the sequestration of CO2 into the stored organic matter is vital for assessing wetland restoration for climate mitigation potential. Yes, it is essential, but it is only part of the story. The paper seems to be to be an incomplete analysis as it ignores the other radiative gases. This team has dealt with the multi-gas problem in other studies, which they cite and say that it has been dealt with other authors (refs 41 – 43). Still, in part, the authors justify their study by arguing that the assessment of CO2 sequestration attributed to restoration needs to be done in the context of the site and wetlands involved. Without including the other GHGs, a conclusion of the role of restoration in mitigating climate cannot be made.

The authors should have the data to do this since they did, at least the CH4, and possibly the N2O EC measurements alongside the CO2 measurements. Vegetation and the same environmental variables that are important for CO2 uptake, are also important, though differently for CH4. In other papers (Environ Res Lett, 2018; 13(4), 045005). This group has shown that ebullition is important in these wetlands. Still, the CH4 production that allows the concentration of CH4 to build up to a level where ebullition can be supported is critically dependent on the vegetation in their study sites. It is the teams ERL paper that provides the argument for the need to analyze the other gases to assess the climate mitigation – “Fifth, as wetlands develop, the relative importance of CO2 vs. CH4 vs. N2O in constraining net GWP may vary significantly,”

The study could be completed by at least adding CH4. What would this do – the authors could make second x-axis on their graphs (Fig 4 & 6) that would have net GHG exchange in CO2 equivalents. Then the conclusions would change substantially. Rather than 2 to 3 years being the critical cross over time, it would be sometime later - one to many decades later, depending on the strength of the CH4 flux. One of the authors has participated in a study that explicitly treats the two- gas problem for wetlands (Proc Natl Acad Sci, 2015; 112(15), 4594-4599). I am not sure if N2O is important – it often is not in wetlands, but since the wetlands being restored were used for grazing, it might be important?

If the authors cannot do the assessment, I suggest they should at least acknowledge that the CO2 sequestration is only part of the restoration - climate mitigation. If they do not have sufficient long-term measurements of CH4 and N2O to do a complete analysis, based on their observations in ERL, they could do some back of the envelope calculations to indicate how much the x-axis would shift in their diagrams when the GHG potential is included. We have struggled with the same problem for peatlands and discussed the GHG mitigation potential for restored peatlands in Nugent et al. ERL 14: (https://iopscience.iop.org/article/10.1088/1748-9326/ab56e6. 2019). I am not pushing this paper on the authors but provide it as an example of how the story change be quite different when the analysis is complete.

Nigel Roulet, McGill University January 2021

6. PLOS authors have the option to publish the peer review history of their article (what does this mean?). If published, this will include your full peer review and any attached files.

Reviewer #1: No

Reviewer #2: No

Reviewer #3: **Yes: **Nigel Roulet

---

## [Author Response · Author response to Decision Letter 0]

25 Feb 2021

Response to Reviewers

We would like to thank the editor and the three reviewers for taking time to review the manuscript and for their insightful and constructive comments. Our responses are under each point and section (and in italic in the Response to Reviewers document) and we hope we were able to make all the appropriate changes and explanations and welcome further questions and feedback.

All line numbers refer to the revised marked-up version of the manuscript where changes to the text can be clearly identified.

Reviewer 1

General comments:

Reviewer #1: This study combines eddy covariance data from five different restored peat-rich wetlands in the San Joaquin delta to evaluate the carbon sink potential of restored wetland as a climate change mitigation option. The authors also compile data on plant cover within the flux footprints to better understand simple indicators of when restored sites will act as carbon sinks. The authors found that wetlands quickly became carbon sinks, but the sink strength varied between years, which they refer to as “fragility”. Overall, the results are sounds and this is a valuable data set used to interrogate management options. Aside from some minor corrections, I suggest that the authors also more clearly demonstrate how fragile the carbon sink function of these restored sites is. They compare these systems to forests, but it would also be worthwhile to place the interannual variability into the context of the interannual variability that would be observed in an undisturbed (or at least less disturbed) marsh system. I also think that making some clearer management recommendations/policy recommendations at the end of the paper could strengthen it overall (i.e., should marsh restoration be considered a climate change mitigation strategy? What does “well-managed” look like/require?).

We thank the reviewer for the helpful comments and highlighting the lack of clarity around our description of C sink fragility. We have rephrased the section on C sink strength and persistence and emphasised how even these strong C sinks can very rapidly become considerable C sources (lines 541-549). In the revisions we explain that because most of the C is stored in the peat and protected by flooding, simply reducing the water level can induce disproportionate C losses both from soil C oxidation, but also reduced C uptake efficiency. This C dynamic differs from non-saturated ecosystems making wetland C sinks particularly susceptible to disturbance even with high vegetation cover and many years after restoration. In some cases, the wetlands can be managed to reduce the impacts of disturbance, but this strongly depends on the design of the wetland making it difficult to list specific measures. However, we have added a few example suggestions. We also show an example where a natural cycle of litter build-up and flushing affected the C uptake strength representing other sources of variability that may be less controllable with management or if possible require significant intervention.

Section 4.4 discussing the C budgets (lines 517 onward) has been edited and now also includes a comparison of our annual C budgets to ranges from other sites classed as undisturbed, although, as you precisely point out, it is very difficult to find truly undisturbed wetlands and even fewer C flux datasets thereof, but it provides some context for the budget ranges presented here, in that these wetlands do appear to be representative of other freshwater marsh C dynamics.

We also expanded the importance of C uptake in terms of GHG budgets based on a previous study (see reference 14). In summary, the high C uptake in these wetlands was sufficient to offset the radiative impacts of the CH4 emissions (N2O fluxes were negligible or even negative), emphasising the importance of C uptake from vegetation dynamics and productivity in wetlands for climate change mitigation. We have added this information in the introduction (lines 58-67) and in the discussion (lines 517-566). Based on that study and the information of this study, we emphasise the importance of wetland management options limiting conditions which reduce plant productivity to maintain a strong C sink. We have added a few management examples applicable to our examples (lines 547-549) and refer to our previous study, which provided a more expansive list of management suggestions including CH4 and N2O emissions.

Specific comments:

1. Line 84: How much does the past management actions affect the post-restoration outcome (i.e., could the same restoration approaches result in different outcomes depending on the management during agricultural use)?

The reasoning for this sentence was to highlight the fact that these wetlands have been fundamentally altered from their original natural state (150 years ago), both from draining and farming practices, but also by the restoration methods used. The wetlands are now below the mean sea level, they are surrounded by levees, and have an artificial pumping system to connect them to the local hydrology. This means they are more similar to a constructed wetland than a restored one and only mimic the natural hydrology and functioning.

However, it is true, that the previous agricultural practices also affected the soil nutrient levels, which is touched on in the Supplementary Information (SI 3), e.g. that sites previously planted with corn/maize had higher P levels than sites used as degraded pastures, the latter of which also took longer for vegetation to fill in. Since only few soil cores were available and only one site had soil data from both before and after restoration (from a pasture), it was difficult to make clear predictions of the impacts, so the limited discussion was moved to the SI. Further soil sampling is underway to supplement the limited data currently available to look into this aspect in more detail, but this may be an example where similar restoration methods yielded different wetland systems. However, the underlying soils of the wetlands do vary from mostly C-rich Rindge muck Histosols to more mineral Gazwell and Scribner Mollisols, so there are still too many different variables interacting to directly compare all components and make definitive conclusions.

2. Line 90-91: Is this really a fair assumption? At least in freshwater inland peatlands these losses can be a very important component of the C budget (Roulet et al 2007, Evans et al 2016).

Again, this is an excellent question which is being addressed at our sites in a separate study. Preliminary data have shown that the eddy covariance measurements agree closely with the C accumulation rates measured from long soil cores indicating little lateral loss, but monthly water sampling at the outflows are still being conducted to quantify the aqueous C loss. The assumption that there is little lateral C loss is further supported by the low flow rate/ long residence times of the water in the wetlands (i.e. 7-10 days in the smallest wetland and longer in the larger ones, see reference 46). The sentence in the text has been changed to reflect this.

3. Line 98: But will those values be included here again for management considerations?

Indeed, including methane and nitrous oxide emissions was initially planned, but the final manuscript became too long so this part of the assessment was then moved to a separate study, which has since been accepted as a peer-reviewed book chapter to be published in Spring 2021 (see refence 14). As mentioned above, the findings of that study are now more specifically mentioned in the introduction and discussion to provide better context and additional insight.

4. Line 271: I think it should be “NEE and GEP during the corresponding…”

Corrected.

5. Lines 320-321: Isn’t the correlation between GEP and NEP confounded since the former is calculated from the latter? Do you really need this correlation to explain the findings?

This is indeed the case and is also highlighted at the start of section 4.4 (lines 493-496). However, this type of plot is often used in studies to compare the C uptake efficiency, hence it was initially included here to allow for visual comparison with other studies (lines 550-556). However, we have now removed Figure 7 and only refer to the slope in the discussion to compare with previous studies.

6. Line 358: I suggest rewording this as something like “Based on soil samples, we found that East End had very high nutrients…”

Done. Also added the mean ± SD in parentheses.

7. Line 377: What “despite”. I would guess that the litter build-up is a good indicator that the site is a C sink (i.e., you are visually seeing the C accumulate in the accumulating litter). I guess, the thought is that the litter is a new source for C release via respiration?

Here, “despite” referred to the fact that the dense litter layer may be shading new growth, hence reducing and/or delaying GEP, but also constitutes a large source of C for aerobic respiration. This sentence has been changed to reflect this.

8. Line 437: I’m not sure you really show the fragility in this section. It’s true that they were C sources in some years, but this was mainly in the early years post-restoration or during known disturbances. There are also risks for other types of C sinks (like forests mentioned in this section), so I think the reasons marshes might be more fragile needs to be more clearly explained in this section, or how you define this interannual variability needs to be reworded.

As mentioned above, further explanations on this have been added (lines 541-549) at the end of the paragraph, which was rewritten to be more concise and better differentiate the C sink strength magnitude and variability and why it is important.

Reviewer 2

Reviewer #2: Summary. Freshwater peatlands store large amounts of carbon in their soils and vegetation, and there is growing interest in using wetland restoration, conservation and management techniques as a climate mitigation strategies. This paper takes advantage of long-term eddy covariance CO2 flux measurements and associated measurements of vegetation community and restoration design to measure CO2 sink strength in 5 wetlands. All 5 wetlands experience similar climate conditions, allowing the authors to explore site-specific controls of CO2 sink. The manuscript shows that vegetation recovery followed 2 distinct trajectories based on restoration design and that most became a CO2 sink after reaching 55% vegetation cover. The manuscript offers additional insights into the temporal variability of sink strength. The manuscript is timely and generally well written. I offer a few comments below.

We thank the reviewer for their constructive comments and for taking time to provide a thorough review, including stylistic feedback to improve the manuscript.

General Comments:

The title is a mouthful. I don’t think it is ‘wrong’ per se, it is just really long. I have to wonder if there is any way to trim it down and still give the reader a sense of the story.

We shortened the title to “Productive wetlands restored for C sequestration quickly become net CO2 sinks with site-level factors driving uptake variability”.

The authors are clear that in the current MS they are focusing on time-series of CO2 fluxes to explore their controls and that they are not attempting to conduct a fully greenhouse gas accounting for these systems (e.g., L96-98; L438-440). This decision makes sense to me, especially considering that radiative forcing has been explored elsewhere, but it does limit the ability of the authors to make a strong case for climate mitigation in these systems and explore their use as ‘natural climate solutions’. Is there is a chance to fold in a bit more about CH4 and N2O fluxes in the discussion to further make the case that there is, indeed, a net climate benefit for restoration of these systems once they become net CO2 sinks? A full greenhouse gas balance is beyond the scope of the paper, but perhaps some broad brushstrokes for context?

We agree with this observation and have included more information from our previous study on the overall climate mitigation potentials of these wetlands (see reference 14) to provide better context for this manuscript, hence the respective sections (1 and 4.4 with lines 55-67 and 517-566) have been expanded and also refer to the wider GHG budgets of these sites and their GHG sink potential. To emphasise the focus on the CO2 component in this manuscript, we instead refer to the wetland’s climate mitigation role as a negative emission technology through C sequestration.

I found the paragraph beginning in L389 to come out of nowhere. This is the first mention of Azolla in the entire manuscript aside from in the legend of Figure 3 (where it looks to cover ~15% in a single site). The authors then give a fair amount of space to Azolla, including conversation around its impacts on CH4 and N2O fluxes, which are not even measured in the current work. I take the point that floating plants and algae are non-negligible contributors to NEE in the study, but this discussion felt really disjointed from the rest of the paper to me.

Azolla and Lemna are now mentioned in the previous paragraph (line 428) and the following paragraph has been shortened as well (lines 432-453).

Minor comments:

1. L17. Consider replacing “litter decomposition rates” with “decomposition”. It isn’t just litter that decays slowly (so does SOM, if litter and SOM are actually different things in a peatland?) and I think the word rate is redundant in this context.

Done.

2. L21. Consider removing “potentials”. You show that restoration impacts C sequestration, not just the potential for this process, right?

Done.

3. L27. Consider removing “and rates”

Done.

4. L31. Can you clarify what you mean by “which most wetlands achieved with vegetation establishment”? Are you saying that most systems become sinks once they hit the 55% threshold?

Correct, most wetlands became on average net C sinks once vegetation coverage exceeded 55%, which was reached after the initial growth years (~2 years). The sentence has been changed for clarity to “… which most wetlands achieved once vegetation was established.”

5. L32. I might add in the equation that you used to define sequestration efficiency (NEP/GPP) here.

Done.

6. L42-44. Estimates of carbon stocks in northern peatland soils have recently been expanded. Might be worth looking at -- Nichols, J.E., Peteet, D.M., 2019. Rapid expansion of northern peatlands and doubled estimate of carbon storage. Nature Geoscience 12, 917–921.

Thank you for the reference suggestion. It has been added and the text updated.

7. L50. I think you’re missing a word here. Perhaps “…uncertainty ABOUT whether….

Added.

8. L62. Can you replace the subject, “This”, with something more concrete? Something like “Exploring these site-specific controls requires….”

Done.

9. L65. I think you can rewrite this without making the authors (“us”) a part of the story. “The eddy covariance method allows for direct measurements of net ecosystem exchange (NEE) over an integrated ecosystem-scale area, and generally assumes that the sources are homogeneous and representative of the whole site.”

Done.

10. L81. Consider removing configurations.

Done.

11. L112-113. I think you need to add a comma after irreversible

Done.

12. L138-139. I think you need to add a comma after 2014. I also think that since there were multiple outbreaks you need to use were here. “In 2014, and to a lesser extent in 2017, there WERE insect infestationS that diminished the green aboveground biomsass.”

Done.

13. Table 1. Can you clarify why the dominant plant species in the Sherman Wetland is NA in this table? There is clearly some vegetation there (58% coverage). What does NA mean here?

NA stands for Not Available, since there are no data available yet from ground surveys of vegetation at a species level from this wetland. To clarify, the assumed dominant species have been added to the table, i.e. Typha and S. acutus (as emergent macrophytes are the largest vegetation class from Fig 3), while the % split shows ‘NA’, which has been added to the table key, to reflect the lack of ground survey confirmation.

14. L156. I would not discuss the eddy covariance methods used to measure CH4 if those data are not going to be considered in this paper.

Done.

15. L177-178. This language is redundant

The sentence has been shortened to ”Although the sites are ideal for eddy covariance with consistent wind directions, large fetch, and little upwind interference, the flux measurements may not be representative of the whole site because of the high spatial heterogeneity.”

16. L263. Can you clarify what is meant by “with vegetation cover >0.1” here? When I look at Figure 4a, it looks like your linear relationship was built with all of the data, including sites with lower vegetation cover. What am I missing here?

This is correct. The 0.1 proportion vegetation cover is the x intercept but has now been removed for clarity.

17. Figure 4. Just a note that you discuss this figure in the order 4a, 4c, 4b, 4d in the text. Do you want to move the panels to align with this order? I am also a bit confused by the units used on the vertical axis in Figure 4b and 4d. In the text, you suggest that you are looking at controls of monthly CUMULATIVE NEE and GEP. Yet, the axis units are PER MONTH. If this is a cumulative sum of multiple months, how can it be a per month number?

The figure panels have been rearranged according to the text sequence. The monthly panels show sums of 30 min fluxes for each month, while the annual plots sum the fluxes for the entire year. The monthly plots were added to more accurately compare the fluxes and vegetation cover measurements during the same time period during the growing season to limit the impact of seasonal differences.

18. L167 (and elsewhere). My understanding is that i.e. (and e.g.) should be followed by commas “i.e.,” as these are abbreviations for phrases.

Both with and without are considered correct, but a ‘following comma’ is more common in American English, as well as when the abbreviation precedes a longer sentence fragment, e.g., as used in this example sentence to demonstrate. After checking the instances in the manuscript, these abbreviations were only used before lists or single words or numbers, so the following comma has been omitted.

19. L269-273. I think this is a run-on sentence. Consider breaking it up and starting a second sentence with “Hence, we only discuss…”

The sentence has been split into “A finer resolution classification showed greater spatial complexity and clear differentiation of vegetation types (Figure S1 in SI). Relationships with monthly and annual NEE and GEP during the corresponding timeframe were only marginally improved compared with the lower resolution images of the same period (Table S1), so only the coarser classification results which cover a longer period are discussed”

20. L358-364. I appreciate the decision to put the detailed nutrient information in the supplement to streamline the story in the main body of the manuscript. However, I wonder if there is a way to be a bit more specific about nutrient levels in the main text. Could you provide summary statistics instead of “very high” and “low” when discussing nutrient levels.

Mean and standard deviations from the supplementary information on nutrient levels have been added to the text (lines 398-400) to provide more accurate descriptions.

21. L396. Replace “methane” with “CH4” – you’ve already used the abbreviation elsewhere

Done.

22. L484. Remove “here”.

Done.

23. L489. Replace “methane” with “CH4”

Done.

Reviewer 3

Reviewer #3: This is a fascinating paper, and I have no comments on the details. The manuscript is well written; the analysis is clear, the methods sound and the conclusions are based on the evidence provided.

However, I have one concern about the manuscript utility. If the goal is to assess the vegetation dynamics through time and how it affects the CO2 sequestration that is fine, this is of little use for wetlands and is only part of the story. This is especially true as the authors' selling point is the importance of knowing how the sequestration of CO2 into the stored organic matter is vital for assessing wetland restoration for climate mitigation potential. Yes, it is essential, but it is only part of the story. The paper seems to be to be an incomplete analysis as it ignores the other radiative gases. This team has dealt with the multi-gas problem in other studies, which they cite and say that it has been dealt with other authors (refs 41 – 43). Still, in part, the authors justify their study by arguing that the assessment of CO2 sequestration attributed to restoration needs to be done in the context of the site and wetlands involved. Without including the other GHGs, a conclusion of the role of restoration in mitigating climate cannot be made.

The authors should have the data to do this since they did, at least the CH4, and possibly the N2O EC measurements alongside the CO2 measurements. Vegetation and the same environmental variables that are important for CO2 uptake, are also important, though differently for CH4. In other papers (Environ Res Lett, 2018; 13(4), 045005). This group has shown that ebullition is important in these wetlands. Still, the CH4 production that allows the concentration of CH4 to build up to a level where ebullition can be supported is critically dependent on the vegetation in their study sites. It is the teams ERL paper that provides the argument for the need to analyze the other gases to assess the climate mitigation – “Fifth, as wetlands develop, the relative importance of CO2 vs. CH4 vs. N2O in constraining net GWP may vary significantly,”

The study could be completed by at least adding CH4. What would this do – the authors could make second x-axis on their graphs (Fig 4 & 6) that would have net GHG exchange in CO2 equivalents. Then the conclusions would change substantially. Rather than 2 to 3 years being the critical cross over time, it would be sometime later - one to many decades later, depending on the strength of the CH4 flux. One of the authors has participated in a study that explicitly treats the two- gas problem for wetlands (Proc Natl Acad Sci, 2015; 112(15), 4594-4599). I am not sure if N2O is important – it often is not in wetlands, but since the wetlands being restored were used for grazing, it might be important?

If the authors cannot do the assessment, I suggest they should at least acknowledge that the CO2 sequestration is only part of the restoration - climate mitigation. If they do not have sufficient long-term measurements of CH4 and N2O to do a complete analysis, based on their observations in ERL, they could do some back of the envelope calculations to indicate how much the x-axis would shift in their diagrams when the GHG potential is included. We have struggled with the same problem for peatlands and discussed the GHG mitigation potential for restored peatlands in Nugent et al. ERL 14: (https://iopscience.iop.org/article/10.1088/1748-9326/ab56e6. 2019). I am not pushing this paper on the authors but provide it as an example of how the story change be quite different when the analysis is complete.

Nigel Roulet, McGill University January 2021

Dear Nigel Roulet,

Thank you for taking the time to review our manuscript and for your helpful feedback and discussion.

We fully agree that vegetation and CO2 uptake are only one aspect of the climate mitigation potential of wetlands and are very glad to see this being highlighted. We recently conducted a more detailed study on exactly this issue (reference 14), which included the CH4 fluxes (and touched on N2O, since there are very few available data), as well as a discussion of different metrics to calculate GHG budgets and how they affect the climate forcing of wetlands. Because of this separate study we specifically did not go into more detail about it in this manuscript. The previous study also looked at the options of reducing CH4 emissions and generally improving the CH4 to CO2 uptake ratio in wetland restoration, which included a table of management suggestions, which we now refer to here. Just to summarise, we found that as expected lower CH4 emissions resulted in smaller GHG budgets. Generally, the wetlands in the Delta have quite high CH4 emissions even under drier conditions (e.g. around 20 gC CH4 m-2 yr-1). However, these sites were not just large C sinks, but we found that the high productivity was enough to offset the radiative impact of the CH4 causing the wetland to be a net GHG sink. In some cases, it was enough to make them an immediate net GHG sink using several GWP conversion metrics, while all wetlands that were net C sinks on average became cumulative net GHG sinks with a crossover time around 50-200 years after restoration. Therefore, at these sites the large net C uptake through the vegetation is key in maintaining their climate mitigation function. Because of that study, we focus here on how vegetation dynamics unfold after restoration and how they relate to restoration design, so that C uptake can be maximised and maintained in the long-term through targeted design and management strategies. So as to better align the reader’s expectations and not overreach the scope of the present study we have rephrased the text to focus on wetlands as negative C emission technologies as part of their climate change mitigation potential. We have added the results of our previous studies in the introduction to better frame the context here (lines 58-67) and also in the discussion in Section 4.4. (lines 516-566).

We generally have come to the conclusion now that we have longer and more complete datasets that freshwater wetlands can be very effective both sequestering C (albeit at the cost of greater short-term radiative forcing), as well as protecting the existing large soil C stocks over geological timescales making them important systems for climate change mitigation. The key nuance is that these properties can easily fluctuate or reverse under certain conditions, hence the call for greater monitoring and management. Furthermore, there is currently much focus on only considering the immediate radiative impacts, which are often lower in other wetland types, such as salt marshes, however, those systems tend to also have much lower productivity and sequester considerably less C in the long-term. Naturally, these systems are also valuable and worth protecting and restoring. Based on our data we aim to highlight that productive wetlands which otherwise may be deemed undesirable because of larger CH4 emissions, should still be a conservation and restoration priority because of their long-term climate change mitigation benefits. With our studies we hope to encourage further discussions about how to accurately estimate the climate impacts of wetlands over climate-relevant timescales and therefore better assess and prioritise restoration/conservation activities. To achieve this, we need to communicate these nuances to policymakers, restoration practitioners, and the wider community.

We apologise that this previous study was not better highlighted in this manuscript. It was supposed to be published in summer 2020 but was delayed due to the virus and now is rescheduled to be published this spring. Because of copyright issues we specifically refrained from reproducing key content in case this paper was published first. We hope the new phrasing, context, and discussion has improved the manuscript’s focus and utility, but would be delighted to discuss this issue further with you.

---

## [Editor Report · Decision Letter 1]

26 Feb 2021

Productive wetlands restored for carbon sequestration quickly become net CO_2_ sinks with site-level factors driving uptake variability

PONE-D-20-35610R1

Dear Dr. Valach,

We’re pleased to inform you that your manuscript has been judged scientifically suitable for publication and will be formally accepted for publication once it meets all outstanding technical requirements.

Kind regards,

Hojeong Kang

Academic Editor

PLOS ONE

Additional Editor Comments (optional):

The authors have properly addressed and clarified all issues that had been raised by the reviewers. The paper is now acceptable for the publication.
---

## [Editor Report · Acceptance letter]

15 Mar 2021

PONE-D-20-35610R1 

Productive wetlands restored for carbon sequestration quickly become net CO_2_ sinks with site-level factors driving uptake variability 

Dear Dr. Valach:

I'm pleased to inform you that your manuscript has been deemed suitable for publication in PLOS ONE. Congratulations! Your manuscript is now with our production department. 

Kind regards, 

on behalf of

Professor Hojeong Kang 

Academic Editor

PLOS ONE